# Let's invert it! From RDF to relational data with R2RML

Arcangelo Massari[1,2,*], Anastasia Dimou[2]

[1]*Research Centre for Open Scholarly Metadata, Department of Classical Philology and Italian Studies, University of Bologna, Bologna, Italy*

[2]*KU Leuven – Flanders Make@KULeuven – Leuven.AI, Department of Computer Science, Leuven, Belgium*

## Abstract

Knowledge graph construction from relational databases relies on mapping languages, such as R2RML, to transform source data into RDF. While the forward transformation is well investigated, the backwards transformation of reconstructing the original data from an RDF graph using the same mapping remains largely unexplored. This paper presents an algorithm for R2RML mapping inversion that generates SPARQL queries from the mapping document to reconstruct the source data, along with a proof-of-concept implementation. We validate the algorithm against the W3C R2RML test suite applied in reverse: all test cases with well-formed and supported mappings are correctly inverted, while the remaining cases, which reveal unsupported features and information loss that prevent reconstruction, are analyzed. Based on these results, we discuss the conditions under which R2RML mappings can be inverted and the structural factors that prevent inversion.

## Keywords

R2RML, knowledge graph construction, mapping invertibility, relational databases

## 1. Introduction

Relational databases remain the most widely adopted data management systems, with the four highest-ranked systems in the DB-Engines popularity index all being relational [1]. The RDB to RDF Mapping Language (R2RML) [2] is a W3C standard for converting relational databases into RDF graphs. In some scenarios, it is useful to preserve both the original relational data and the derived RDF graph, as they serve different purposes: the relational representation supports existing applications and workflows, while the knowledge graph facilitates data sharing according to the FAIR principles [3]. When both representations coexist, keeping them synchronized requires the ability to reconstruct relational data from the RDF graph, yet this backwards transformation has received limited attention.

This paper presents an algorithm for inverting R2RML mappings that generates SPARQL queries from R2RML mapping documents to reconstruct the original relational data from RDF graphs. A proof-of-concept implementation is available as open-source software under the ISC license [4]. The algorithm is validated against the W3C R2RML test suite [5]: each test case provides source data, an R2RML mapping, and expected RDF output; the algorithm attempts to reconstruct the original input from the RDF output using the same mapping. The results are used to discuss the conditions under which inversion succeeds and the structural factors that prevent it.

## 2. Related work

Two works have been proposed so far to reconstruct raw data from RDF: RML2CSV and the mapping-template tool. **RML2CSV** [6] was introduced to reverse RDF datasets back into their original CSV structures using the same mappings that generated the RDF graphs. However, this algorithm relies on the Dependency Tree Assumption, requiring mappings to form a single n-ary tree with clear parent-child relationships, which restricts its use to strictly hierarchical data structures. Additionally, it

*7th International Workshop on Knowledge Graph Construction (KGCW 2026), co-located with ESWC 2026, May 10–11, 2026, Dubrovnik, Croatia*

*Corresponding author.

✉ arcangelo.massari@unibo.it (A. Massari); anastasia.dimou@kuleuven.be (A. Dimou)

🆔 0000-0002-8420-0696 (A. Massari); 0000-0003-2138-7972 (A. Dimou)

assumes implicit cardinality constraints of 0:0 or 1:1 between CSV columns, making it inadequate for handling one-to-many or many-to-many relationships. The **mapping-template tool** [7] addresses knowledge conversion from RDF to arbitrary output formats using the Mapping Template Language (MTL), built on the Apache Velocity Engine. This approach supports the reverse direction (RDF to non-RDF) without requiring knowledge of RDF from the mapping author. However, it requires writing dedicated templates for the output direction, independent of any existing R2RML mappings used for the forward transformation. In contrast, our approach reuses the same R2RML mappings for both directions, generating SPARQL queries directly from the mapping specification without additional authoring effort.

## 3. Inversion algorithm

The starting point is a relational database table, an R2RML mapping that was used to transform the table into RDF, and the resulting RDF graph. The goal of our algorithm is to reconstruct the original relational data from the RDF graph using only the mapping document, without access to the original database. The algorithm assumes that the RDF graph is the direct output of the forward transformation and has not been subsequently modified.

Since the RDF graph is the only data source available for reconstruction, SPARQL provides the extraction mechanism. The mapping document encodes how each column value was transformed into RDF terms during the forward process. The algorithm reads this encoding and generates SPARQL queries that reverse it. For example, where the forward transformation embedded a column value into an IRI through a template, the SPARQL query extracts it back through string manipulation.

The algorithm proceeds in three stages: (i) it **extracts the source table name** from the logical source specification in the mapping; then (ii) it inspects all term maps to discover **which columns** the table contains and how their values were encoded into RDF; and last, (iii) it assembles a SPARQL query and executes it against the RDF graph to **extract each column's value** and reconstructs the table rows.

The following sections describe the inversion strategies for different term map types, using a running example drawn from the W3C R2RML test suite [5]. Table 1 shows the source data, and Listing 1 shows the corresponding R2RML mapping.

**Table 1**
Source table "Student" (from R2RMLTC0007a).

| ID | Name |
|----|------|
| 10 | Venus |

Listing 1: R2RML mapping for the student table.

```
1  @prefix rr: <http://www.w3.org/ns/r2rml#> .
2  @prefix foaf: <http://xmlns.com/foaf/0.1/> .
3  @base <http://example.com/base/> .
4
5  <TriplesMap1>
6      a rr:TriplesMap;
7      rr:logicalTable [ rr:tableName "\"Student\"" ];
8      rr:subjectMap [ rr:template
9        "http://example.com/Student/{\"ID\"}/{\"Name\"}" ];
10     rr:predicateObjectMap [
11       rr:predicate rdf:type ;
12       rr:object foaf:Person ] .
```

### 3.1. Column discovery

The algorithm identifies which columns need to be reconstructed by inspecting all term maps in the triples map: the subject map, predicate maps, object maps, and graph maps. Each term map that references a column, either through `rr:column` or through a placeholder in `rr:template`, corresponds to a column in the source table. In the running example, the subject template `http://example.com/Student/{ID}/{Name}` references two columns: `ID` and `Name`.

Three types of term maps exist in R2RML, each requiring a different inversion strategy.

**Constant-valued term maps** use `rr:constant` for subjects, predicates, or objects. Since constants do not reference any column, they filter the SPARQL results without contributing to column reconstruction. In Listing 1, `rr:predicate rdf:type` (line 11) and `rr:object foaf:Person` (line 12) are both constants: the generated SPARQL in Listing 2 (line 2) embeds them as a fixed triple pattern `?Name_uri a <.../Person>`, which selects subjects by type without extracting any column value.

**Reference-valued term maps** use `rr:column` to reference a source column directly. The column value appears in the RDF graph as a literal (default) or as an IRI (when `rr:termType rr:IRI` is specified), so the SPARQL query captures it with a variable that binds to the cell value without string manipulation. For example, if a mapping specifies `rr:objectMap [ rr:column "Name" ]`, the pattern `?s foaf:name ?Name` binds `?Name` directly to the column value. The same applies when a column reference is used as the subject with `rr:termType rr:IRI`: the subject IRI is the column value itself, so no STRAFTER/STRBEFORE extraction is needed.

**Template-based term maps** encode column values within IRI structures using `rr:template`. Reconstructing the original values requires string manipulation to reverse the encoding, as described in the next section.

The algorithm assumes that each column is referenced only once across the term maps of a triples map. If the same column appeared in multiple term maps, the algorithm would need to determine which one to use for extraction and verify that all references produce consistent values. In some cases one term map might generate a term while another does not: for instance, a column value that is valid as a literal but causes the IRI generation to fail in a template-based term map would be recoverable from one path but not the other.

### 3.2. Template inversion

The subject template in Listing 1 is `http://example.com/Student/{ID}/{Name}`, which encodes both column values into the IRI. The forward transformation produces `<http://example.com/Student/10/Venus>`. To recover the column values, the algorithm generates the SPARQL query in Listing 2. The `FILTER(REGEX(...))` on line 3 selects subjects matching the template structure. Sequential STRAFTER/STRBEFORE calls then extract each column value from left to right: STRAFTER on line 5 removes the prefix up to `.../Student/`, yielding `"10/Venus"`; STRBEFORE on line 7 isolates `"10"` as the ID; STRAFTER on line 9 captures `"Venus"` as the Name. When a column encoded in the subject template also appears as a literal in a predicate-object map, the algorithm uses the literal value directly and skips the template extraction for that column.

Listing 2: SPARQL query generated for the student table inversion.

```
1   SELECT ?Name ?ID WHERE {
2     ?Name_uri a <http://xmlns.com/foaf/0.1/Person> .
3     FILTER(REGEX(STR(?Name_uri),
4       '.../Student/([^/]*)/([^/]*)'))
5     BIND(STRAFTER(STR(?Name_uri),
6       '.../Student/') AS ?Name_uri_slice)
7     BIND(STRBEFORE(STR(?Name_uri_slice),
8       '/') AS ?ID)
9     BIND(STRAFTER(STR(?Name_uri_slice),
10      '/') AS ?Name)
11  }
```

This approach generalizes to any template with $n$ placeholders separated by literal segments. The algorithm converts the template into a regular expression by replacing each placeholder with a capture group (`[^/]*`), applies a `FILTER(REGEX(...))` to select matching subjects, and extracts values left to right with alternating `STRAFTER`/`STRBEFORE` calls. The extraction relies on these literal segments to determine where one column value ends and the next begins; when placeholders are concatenated without separators (e.g., `{fname}{sname}`), the boundary is indeterminate and the inversion fails. For instance, given the IRI segment `johnsmith`, there is no way to determine where the first name ends and the surname begins.

### 3.3. Referencing object maps

Predicate maps and object maps follow the same constant, reference, and template strategies. Object maps additionally support referencing object maps, which link two triples maps through `rr:parentTriplesMap` and `rr:joinCondition` to represent relationships between tables. Inverting this relationship requires extracting the child column value from the parent subject IRI. Table 2 shows an example from R2RMLTC0009a: the "Student" table references the "Sport" table through a foreign key column. The mapping (Listing 3) links the two triples maps via a join condition matching the child column "Sport" against the parent column "ID".

**Table 2**
Source tables with foreign key relationship (from R2RMLTC0009a).

| | Student | | | Sport | |
|----|------|-------|--|------|------|
| ID | Name | Sport | | ID | Name |
| 10 | Venus Williams | 100 | | 100 | Tennis |
| 20 | Demi Moore | NULL | | | |

Listing 3: R2RML mapping excerpt with referencing object map.

```
1  <TriplesMap1>
2      rr:logicalTable [ rr:tableName "\"Student\"" ];
3      rr:subjectMap [ rr:template
4        "http://example.com/resource/student_{\"ID\"}" ];
5      rr:predicateObjectMap [
6        rr:predicate foaf:name ;
7        rr:objectMap [ rr:column "\"Name\"" ] ];
8      rr:predicateObjectMap [
9        rr:predicate ex:practises ;
10       rr:objectMap [
11         rr:parentTriplesMap <TriplesMap2>;
12         rr:joinCondition [
13           rr:child "\"Sport\"" ;
14           rr:parent "\"ID\"" ] ] ] .
15
16 <TriplesMap2>
17     rr:logicalTable [ rr:tableName "\"Sport\"" ];
18     rr:subjectMap [ rr:template
19       "http://example.com/resource/sport_{\"ID\"}" ];
20     rr:predicateObjectMap [
21       rr:predicate rdfs:label ;
22       rr:objectMap [ rr:column "\"Name\"" ] ] .
```

The forward transformation produces four triples: `student_10` has a name and practises `sport_100`, `student_20` has a name but no `practises` triple (the NULL foreign key generates no triple), and `sport_100` has a label. To invert the student table, the algorithm generates the query in Listing 4. Line 2 retrieves the literal name directly. The OPTIONAL block on lines 3–7 handles the foreign key:

the value "100" is extracted from the parent subject IRI `sport_100` via STRAFTER (line 5). Since Demi Moore has no `practises` triple, the `?Sport` variable remains unbound, which the algorithm writes back as NULL. Lines 8–11 filter subjects matching the student template and extract the ID value.

Figure 1 shows the flowchart for term map processing. Subject, predicate, and graph maps follow the constant, reference, and template branches; the referencing object map branch applies only to object maps.

Listing 4: SPARQL query for inverting the student table with foreign key.

```
1   SELECT ?Name ?Sport ?ID WHERE {
2     ?student__ID_uri foaf:name ?Name .
3     OPTIONAL { ?student__ID_uri
4         ex:practises ?sport__ID_uri .
5       BIND(STRAFTER(STR(?sport__ID_uri),
6         '.../resource/sport_') AS ?join_slice)
7       BIND(?join_slice AS ?Sport) }
8     FILTER(REGEX(STR(?student__ID_uri),
9       '.../resource/student_([^/]*)'))
10    BIND(STRAFTER(STR(?student__ID_uri),
11      '.../resource/student_') AS ?ID) }
```

### 3.4. Graph maps

R2RML allows assigning generated triples to named graphs through `rr:graphMap`, which can use the same constant, reference, and template types as other term maps. A **constant-valued graph map** (e.g., `rr:graph ex:PersonGraph`) does not reference any column and has no effect on reconstruction: the triples are simply retrieved from the named graph. A **template- or reference-valued graph map** encodes column values in the graph IRI. When these columns also appear in other term maps (subject, predicate, or object), the algorithm recovers them from there without additional extraction. When a column is referenced exclusively by the graph map, the algorithm wraps the triple patterns in a GRAPH `?g { ... }` clause and applies the same STRAFTER/STRBEFORE extraction to the graph variable.

## 4. Validation

The algorithm was validated using the W3C R2RML test suite [5], which consists of 62 test cases covering a range of mapping scenarios for relational databases. Each test case includes source data, a mapping document, and expected RDF output. The test suite was applied in reverse: the algorithm reconstructs the original data from the RDF output using the same mapping, then compares the result with the original input. PostgreSQL was used as the database system.

Out of the 62 test cases, 17 involve SQL queries as logical sources (`rr:sqlQuery`), which the current algorithm does not handle. Excluding these, **45 test cases** are relevant for validation. Table 3 summarizes the results. The **26 successfully inverted test cases** cover template-based subject and literal construction, named graphs, SQL datatype preservation, relationship reconstruction through referencing object maps and join conditions, and column-as-IRI subjects as described in Section 3.

The **16 non-invertible cases** expose scenarios where the mapping does not preserve enough information for complete reconstruction. The most frequent case is partial mappings: columns present in the source table but absent from the mapping have no RDF representation. For instance, in R2RMLTC0016a, a "Patient" table with ten columns is mapped using only four (`ID`, `FirstName`, `LastName`, `Sex`). This is not a mapping deficiency but a conscious choice to select a subset of columns for the knowledge graph. The algorithm can still partially reconstruct the table, recovering the mapped columns.

Non-unique subject templates cause multiple source rows to produce the same IRI, collapsing them into a single RDF subject. In R2RMLTC0005a, an "IOUs" table contains two identical rows (`Bob`, `Smith`,

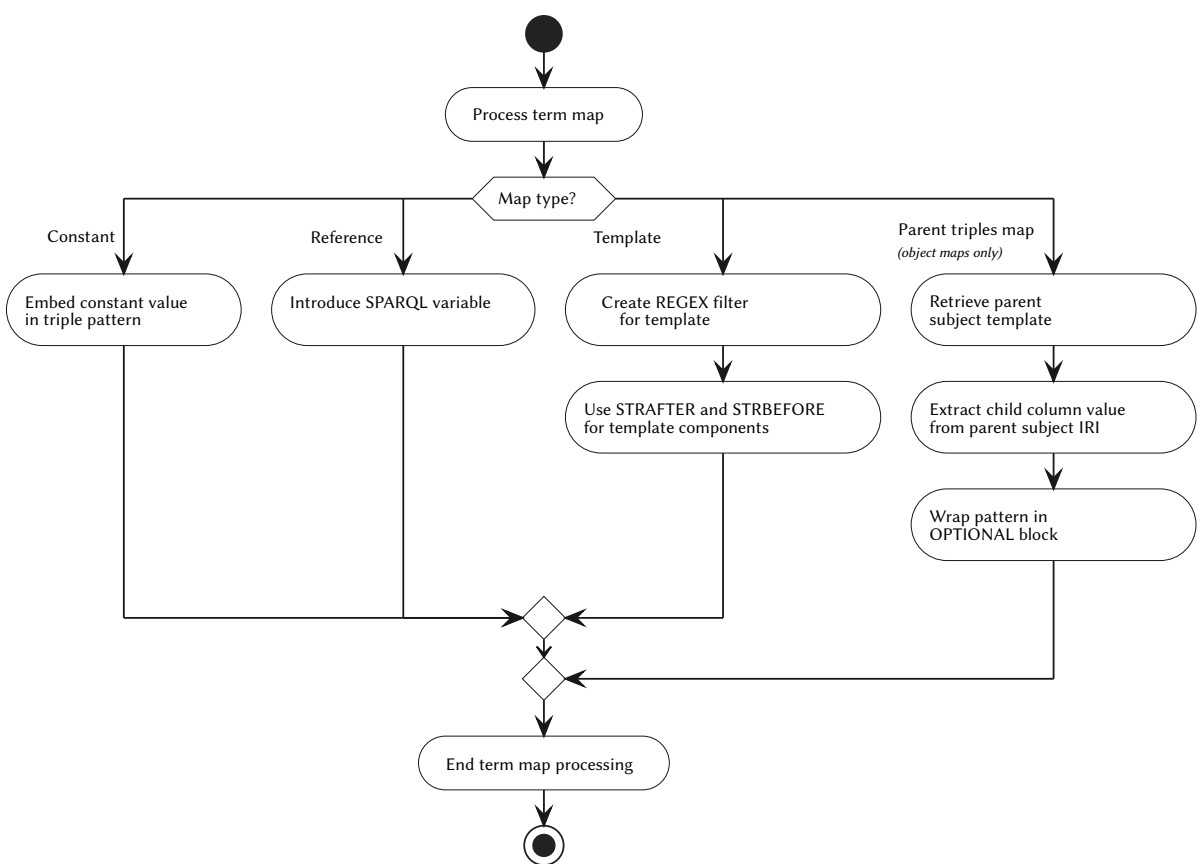

**Figure 1:** Flowchart of term map processing.

**Table 3**
W3C R2RML test suite validation results (45 test cases, excluding SQL query logical sources).

| Category | Sub-category | Count |
|---|---|---|
| Successfully inverted | | 26 |
| Non-invertible cases | Partial mappings | 9 |
| | Non-unique subject templates | 3 |
| | Combined cases | 2 |
| | Constant-only mapping | 1 |
| | NULL in subject template | 1 |
| | *Subtotal* | *16* |
| Invalid mappings (correctly rejected) | | 3 |

30); the subject template `http://example.com/{fname};{lname}` maps both to the same IRI, and the resulting identical triples merge into one.

Two test cases exhibit combined cases: one has an entirely unmapped source table together with non-unique subjects, and another combines partial mappings with non-unique subjects across different tables.

A constant-only mapping produces triples where every component is a fixed value with no column references. In R2RMLTC0006a, the mapping produces ex:`BadStudent` ex:`description` "Bad Student" regardless of the table content. Since no term map references any column, the mapping is indifferent to the source data and reconstruction does not apply.

NULL values in columns referenced by the subject template exclude the corresponding rows from the RDF output entirely, as R2RML specifies that NULL template references produce no triples. In R2RMLTC0013a, a "Person" table includes a row where the `DateOfBirth` column is NULL; since the

subject template `http://example.com/Person/{ID}/{Name}/{DateOfBirth}` references this column, the row generates no triples.

The 3 invalid mapping cases were correctly detected and rejected: a mapping using literals as graph names, a mapping lacking a required subject map, and a mapping with multiple subject maps per triples map.

## 5. Discussion and conclusion

The validation results show that R2RML mapping inversion is feasible but not universally applicable. All test cases where the mapping references a source table and preserves column information are correctly inverted. The cases where inversion fails can be classified into categories that reflect structural characteristics of the mapping rather than algorithmic limitations:

(i) **coverage**. When a mapping selects only a subset of columns for the knowledge graph, the unmapped columns have no RDF representation and cannot be reconstructed. This is a conscious design choice: the algorithm can still partially reconstruct the table, recovering the mapped columns.

(ii) **subject uniqueness**. When a subject template produces the same IRI for multiple source rows, those rows collapse into a single RDF subject and the original row count is lost. This represents an irreversible transformation regardless of the inversion approach.

(iii) **template structure**. The STRAFTER/STRBEFORE extraction relies on fixed literal segments separating the placeholders in the template. For literal templates (where R2RML does not apply URL-encoding), separator characters appearing within column values produce ambiguous extractions: the template `{FirstName} {LastName}` with a first name containing a space (e.g., "Mary Jane") causes incorrect splitting. IRI templates avoid this because R2RML mandates URL-encoding of column values. As described in Section 3, templates with concatenated placeholders without separators also prevent inversion. The assumption on single column references described in Section 3 further constrains the cases where template inversion succeeds: if a column appears in multiple term maps, the algorithm would need to reconcile potentially inconsistent extraction results.

(iv) **NULL propagation**. NULL values in columns referenced by the subject template prevent the entire row from generating any triples, making that row invisible in the graph. NULL values in columns referenced only by object maps omit individual triples; if other triples exist for the same subject, the algorithm reconstructs the missing values as NULL, but if no other triples remain, the row is equally lost.

This paper does not aim to be exhaustive and complete, but it seeks to initiate the discussion and hopefully foster further research in this direction. Several aspects remain outside the scope of this work. Logical sources with SQL queries (`rr:sqlQuery`) were excluded from the validation; inverting arbitrary SQL with joins, aggregations, and subqueries is a distinct problem that may not admit a general solution in all cases. Blank node handling depends on factors outside the mapping specification: R2RML requires templates for blank node generation, so the same string extraction could apply in principle, but triplestores and RDF libraries typically replace blank node identifiers with opaque internal identifiers. Since R2RML is the relational subset of the RDF Mapping Language (RML) [8, 9], these results apply to the relational component of RML. Extending the inversion to non-relational sources supported by RML, which also allows random blank nodes without templates, remains an open question.

It also remains an open question how the reverse mapping could be extended for RML in general. What are the additional complications? How can we reconstruct the schema of hierarchical data sources? What about lists? Other future directions include investigating the feasibility of inverting (SQL) view logical sources, evaluating the approach on real-world mappings beyond the W3C test suite, and assessing scalability on large RDF datasets.

## Acknowledgments

The authors thank Mario Scrocca (ORCID: 0000-0002-8235-7331) for sharing his experience on the

mapping inversion problem.

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
