# OpenReview forum: "Let’s invert it! From RDF to relational data with R2RML"
_eswc-conferences.org/ESWC/2026/Workshop/KGCW — KGCW 2026_

### Official Review · ~Franck_Michel1 · 2026-04-02
**Easy-to-read paper addressing an unexplored need**

**Rating:** 7
**Confidence:** 5

**Review:**

This paper proposes a method to reconstruct relational tables from an R2RML mapping graph and the resulting graph that was generated from the relational data. In addition to the method, it points to an open source PoC implementation, and provides the results of an evaluation carried out using the R2RML test cases.
The paper also clearly outlines the limitations of the approach, which do not result from the method itself but from information that cannot be recovered simply based on the R2RML mapping and the produced graph.
The writing style is very clear, and the whole paper is easy to follow.

A possible limitation that is not mentioned is with respect to the handling of datatypes. R2R2ML maps SQL datatypes to XSD types in a many-to-one manner, such that recovering the source datatype may not be possible. For instance, SMALLINT, INTEGER and BIGINT are mapped to xsd:integer. Can this be a problem when reverting a literal with type xsd:integer to its original SQL value? Or is this transparent in the parsing? This should be touched upon in the paper.
The approach is motivated by the need, in some contexts, to keep the RDB and the graph synchronized, which requires the ability to reconstruct relational data from the RDF graph. Such a synchronization approach should typically focus on deltas (reconstruct only the data that has changed since a certain time). Although this is out of the scope of the paper, the conclusion may touch upon this because maintaining change logs in triple store remains an open question.

Typos:
Listings 1 and 3 contain unneeded “\”-escaped double-quotes.
“in contrast, …”=> ‘”by contrast, …”

---

### Official Review · ~Michael_Freund1 · 2026-04-02
**Interesting Idea, but Motivation and Scope Could Be Clearer**

**Rating:** 6
**Confidence:** 5

**Review:**

The paper addresses the inversion of R2RML mappings with the aim of reconstructing relational source data from RDF graphs using the original mapping specification. This is a relevant and relatively unexplored topic, and the idea of using the same mapping for forward and backward transformations is an interesting contribution. Validation using the W3C R2RML test suite also provides a clear basis for the evaluation.

The paper is easy to follow and presents a promising proof of concept. At the same time, some parts could be clearer. In particular, the motivation could be strengthened by discussing the practical scenarios in which inversion is needed in a bit more explicit detail. Additionally, the notion of invertibility should be defined more precisely, as it is unclear whether inversion refers to the exact reconstruction of the original relational data (including rows, columns, null values and duplicates) or the recovery of all available information after the forward transformation. This distinction is important for interpreting the results, particularly in cases involving partial mappings, non-unique subject templates and null values in subject generation, where some information may still be recoverable even if exact reconstruction is no longer possible.

Overall, I find the paper to be a relevant contribution to the workshop, particularly given that it initiates discussion on an issue that has received limited attention.

---

### Official Review · ~Oscar_Corcho1 · 2026-04-06
**Interesting approach on a topic that has received little attention**

**Rating:** 5
**Confidence:** 4

**Review:**

The possibility of reconstructing original relational databases from RDF that has been generated using R2RML has received very little attention in the state of the art, mostly due to the fact that the proposal for R2RML (as well as other mapping languages) was focused exclusively on the forward translation that was needed at that stage, and the little value that such reconstruction may bring in.

As such, the paper shows an interesting discussion on what is possible and what is not possible, exemplified by the R2RML test cases, where the analysis already shows that a large number of them do not allow for a reconstruction (that is, for the inverse transformation). This situation would be exacerbated if we would go for additional features that are present in RML and its extensions (functions that are not necessarily bijective, other types of transformations, etc.)

The reconstruction process is well presented and discussed, with a good characterisation of types of mappings, and the method that is proposed seems correct and uses well-known languages such as SPARQL.

---

### Decision · Program_Chairs · 2026-04-09

**Decision:**

Accept

**Comment:**

This paper has been selected for presentation at the KGC workshop. We strongly encourage the authors to consider the reviews whilst revising the paper. Camera-ready instructions will soon follow.